# Using LLM for Improving Key Event Discovery: Temporal-Guided News Stream Clustering with Event Summaries

**Nishanth Nakshatri**♣    **Siyi Liu**◇    **Sihao Chen**◇
**Daniel J. Hopkins**◇    **Dan Roth**◇    **Dan Goldwasser**♣

♣Purdue University ◇University of Pennsylvania
{nnakshat,dgoldwas}@purdue.edu
{siyiliu, sihaoc, danhop, danroth}@upenn.edu

## Abstract

Understanding and characterizing the discussions around key events in news streams is important for analyzing political discourse. In this work, we study the problem of identification of such key events and the news articles associated with those events from news streams. We propose a generic framework for news stream clustering that analyzes the temporal trend of news articles to automatically extract the underlying key news events that draw significant media attention. We characterize such key events by generating event summaries, based on which we form document clusters in an unsupervised fashion. We evaluate our simple yet effective framework, and show that it produces more coherent event-focused clusters. To demonstrate the utility of our approach, and facilitate future research along the line, we use our framework to construct KEYEVENTS[1], a dataset of $40k$ articles with 611 key events from 11 topics.

## 1 Introduction

Analyzing the dynamics of discussions within the stream of news coverage has been an important tool for researchers to visualize and characterize media discourse around a topic (Field et al., 2018; Liu et al., 2019; Li and Goldwasser, 2019; Roy and Goldwasser, 2020; Luo et al., 2020; Liu et al., 2021; Lei et al., 2022; Dutta et al., 2022). News media discourse is typically centered around real-world *events* that catch media attention and gives rise to news reports streams. With the vast, ever-growing amount of news information available, we need automatic ways for identifying such key events.

In this paper, we study the problem of identifying and characterizing *key events* from a large collection of news articles. Since the number of news events is usually not known in advance, past works have typically formulated the problem as a form of non-parametric clustering of news articles, using Hierarchical Dirichlet Processes (Zhou et al., 2015; Beykikhoshk et al., 2018) or Stream Clustering (Laban and Hearst, 2017; Miranda et al., 2018; Staykovski et al., 2019; Saravanakumar et al., 2021). Rather than relying on the output of such clustering algorithms directly, we view the discovered clusters as *event candidates*, and leverage recent advances in Large Language Modeling (LLM) (Brown et al., 2020) to characterize these candidates and reason about their validity. From a bird's eye view, the process is related to past work on interactive clustering (Hu et al., 2014; Pacheco et al., 2022, 2023), but instead of using human feedback to shape the emergent clusters, we rely on LLM inference.

We propose a framework for clustering an archive of news articles into temporally motivated news events. A high-level overview of our approach is shown in Figure 1. We first retrieve relevant issue-specific articles (details about the document retrieval module are in App A) and perform temporal analysis to identify "peaks", in which the number of articles is significantly higher. We then use HDBSCAN (Campello et al., 2013) a non-parametric clustering algorithm to generate candidate event clusters. We then *characterize* the candidate clusters by performing few-shot multi-document summarization of the top-K articles assigned to each cluster, identify *inconsistent clusters* by assessing the (dis)agreement between the summary and each article individually, and *redundant clusters* by assessing the similarity between cluster pairs' summaries (details in Sec. 2.1). These low-quality candidates are removed, resulting in higher quality event clusters. We demonstrate this property over the NELA dataset (Horne et al., 2022) and show the improvement both in terms of event coherence and document mapping quality.

---

[1] https://github.com/nnakshat/KeyEvents

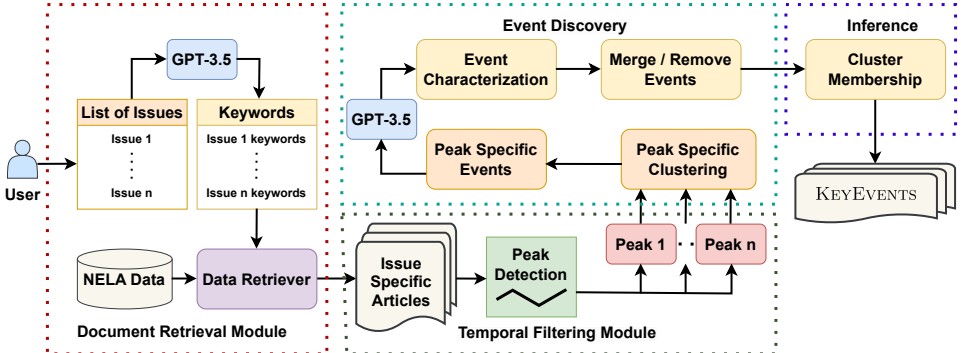

Figure 1: High-level overview of our framework for KEYEVENTS identification.

## 2 Event Discovery and Article Inference

### 2.1 Event Discovery

**Temporal Filtering.** The first step towards generating event candidates is to identify *temporal landmarks* or *peaks*, where the media coverage surges with respect to one or more real-world events. We represent the news articles as a time-series data, where $\mathcal{T} = \{t_1, t_2, \cdots, t_n\}$ denote time, and $\mathcal{C} = \{c_{t_1}, c_{t_2}, \cdots, c_{t_n}\}$ denote the number of articles published at each time step. The task is to identify a set of peaks, $\mathcal{P} = \{p_1, p_2, \cdots, p_m\}$ at different points in time. With this formulation, we hypothesize that the resulting clusters from our framework would be able to segregate discussions at various time steps and form coherent events compared to other approaches. We use an existing *outlier* detection algorithm (Palshikar et al., 2009) towards this task. More details in Appendix B.

**Peak-Specific Clustering.** Within each peak, the increased media coverage can be attributed to multiple relevant events. We categorize the documents in each peak $p_i$ into a set of events, $\mathcal{E}_i = \{e_1, e_2, \cdots, e_q\}$, and form an overall event set, $\mathcal{E} = \{\mathcal{E}_1, \mathcal{E}_2, \cdots, \mathcal{E}_m\}$, pertaining to the issue. We embed the *title* and *first 4 lines* of a news article instance using a dense retriever (Ni et al., 2021) model. The embedded documents are clustered using HDBSCAN to identify key news events. Prior to clustering, we reduce the dimensions of document embedding using UMAP (McInnes et al., 2018). Details are in Appendix C.

**Event Characterization.** The event set obtained at each peak ($\mathcal{E}_i$), is still prone to noise and is not easily interpretable without significant effort. Characterizing the news events makes the clusters interpretable and helps remove inconsistencies. The candidate events are characterized by generating

| Incoherent Cluster (Top-3 documents shown) |
|---|
| **Event Title**: Climate Justice and African Activists
**Event Description**: This is about the challenges faced by African climate activists in bringing attention to the climate crisis and the need for climate justice. |
| **Doc. 1:** *There Will Never Be Climate Justice If African Activists Keep Being Ignored*
We go to Kampala, Uganda, to speak to climate activist Vanessa Nakate on the occasion of her first book being published, A Bigger Picture. ... |
| **Doc. 2:** *The Looking Glass World Of 'Climate Injustice'*
In our wacky world where almost nothing makes sense anymore, there is no shortage of examples of politicians, let alone self-important academics, journalists, and wealthy elites, looking foolish with self-contradictory policy demands. ... |
| **Doc. 3:** *New Miss Universe Urges Action on Climate Change: Choice to Kill or Save Nature*
A new Miss Universe has been crowned and she is a climate alarmist. ... |

Table 1: Incoherent cluster removal. The cluster summary aligns with the $1^{st}$ and the $2^{nd}$ article, while the $3^{rd}$ article is off-topic compared to the other two.

a multi-document summary using GPT-3.5. The prompts are engineered to generate short event-specific summaries in a two-shot setting. The two closest documents to each centroid are used in the prompt to generate event summaries.

Post summary generation, we perform a *cluster inconsistency check*. A cluster is deemed to be incoherent if the top-K closest documents to the centroid do not align with the summary embedding. We embed the event summaries using the same dense retriever model, and compute the cosine similarity score between the summary embedding and the top-K documents for the cluster ($k = 5$). Based on a threshold value, we treat the incoherent clusters as noise and discard them. Note that we only discard clusters but not documents associated with them. They are still used for cluster membership assignment in the next stage of our framework. Tab. 1

| Summary of Article 1 | Summary of Article 2 |
|---|---|
| **Event Title**: President Biden's Climate Plan
**Event Description**: This is about President Joe Biden's executive orders aimed at tackling climate change by reducing the U.S. carbon footprint and emissions, stopping oil and gas leases on public lands, and prioritizing climate change as a national security concern. | **Event Title**: Biden's Climate Change Actions
**Event Description:** This is about President Joe Biden's executive actions to combat climate change by prioritizing science and evidence-based policy across federal agencies, pausing oil drilling on public lands, and aiming to cut oil, gas, and coal emissions. |
| **Event Title**: Texas Abortion Ban
**Event Description**: This is about a new Texas law that bans abortions after 6 weeks and empowers regular citizens to bring civil lawsuits against anyone who aids a woman looking to terminate a pregnancy. | **Event Title**: Texas Abortion Law
**Event Description**: This is about the controversial Texas abortion law that bans abortions after six weeks and has been condemned by President Joe Biden as an unprecedented assault on women's rights. |

Table 2: Illustrates two cases of cluster merge from issue *Climate Change*, and *Abortion* respectively.

shows an example of the discarded cluster.

We do an additional cleaning step by *merging the clusters that share a similar event summary*. We devise a simple greedy algorithm which utilizes GPT-3.5 for inference. In the first iteration of the algorithm, we start by constructing a set, $S = \{(s_1, s_2), \cdots, (s_{n-1}, s_n)\}$, that contains every pairwise combination of event summaries. For each element in $S$, we prompt LLM to infer if the pair of summaries are discussing about the same event. If the event summaries, say $(s_1, s_2)$, are equivalent, then we merge these summaries, and update the set $S$ by removing every element in the set that contains $s_1$ or $s_2$. In the second iteration, we construct a new set, $S'$, that holds every combination of updated event summaries, and repeat the previous step. We run the algorithm for two iterations or halt if there are no merges after the first iteration. Tab. 2 shows an example where the event summaries clearly indicate that the clusters need to merged. Details about the hyperparameter selections, and prompts are in Appendix C, B.

## 2.2 Inference: Map Articles to Events

In this stage of our framework, we decide the cluster membership using a similarity module. We embed the updated event summaries using the same encoder, and compute the cosine similarity score between the summary and the document of interest. By thresholding, we determine if the article can be mapped to an event. For cluster membership, we extend the temporal window by $d$ days before and after the peak ($d = 1$), and consider all the documents published in that timeframe.

## 3 Experiments and Results

We conduct experiments on the NELA-dataset, which is a large collection of news articles (see Appendix A). Using our document retrieval module, we collect a total of $335k$ relevant news articles on 11 contemporary issues[2]. The application of temporal filters reduces the article count to $90k$, which is the basis for our analysis. The retrieved articles are mapped to a four-way {*left*, *right*, *center*, and *conspiracy-pseudoscience*} political rating. Details about the dataset, document retrieval module, and four-way political rating can be found in Appendix A.

**Evaluation Metrics.** We evaluate our framework's ability to create coherent event clusters at the desired granularity with three automatic metrics inspired by Mimno et al. (2011). Given an event $e_i$ and the top-10 relevant entities $V^{e_i} = \{v_l^{e_i}\}_{l \in [1..10]}$ to $e_i$ by TF-IDF, *entity purity* measures the percentage of the documents that mention at least one of the top-10 entities; *coverage* counts the percentage of documents accounted for in the cluster assignments. In addition, *entity coherence* considers co-occurrences of central entity pairs in the clustered documents to measure coherency for an event.

$$C(e_i, V^{e_i}) = \sum_{m=2}^{M} \sum_{l=1}^{m-1} \log \frac{F(v_m^{e_i}, v_l^{e_i}) + \epsilon}{F(v_l^{e_i})}$$

where $F(v_m^{e_i}, v_l^{e_i})$ indicates the co-occurrence frequency of two entities in documents. An entity coherence value closer to zero indicates a highly coherent news event cluster. We offer a more detailed explanation of the metrics in Appendix D.

**Baselines.** We compare our method's performance against various competitive topic model as baselines. We consider LDA (Blei et al., 2003; Hoffman et al., 2010) in two different settings - *LDA*, and *LDA (Temporal)*. The topics are estimated individually at each temporal peak for *LDA (Temporal)*, whereas the topics are estimated across

[2]https://www.allsides.com/topics-issues

| Model | Coverage↓ | Entity Purity↑ | Entity Coherence↑ | Event Count |
|---|---|---|---|---|
| LDA (baseline) | 99.69 | 31.52 | -1008.42 | 60.0 |
| Temporal filtering | - | 28.15 | -1061.60 | 18.7 |
| LDA (Temporal) | 89.02 | 38.62 | -1005.37 | 65.7 |
| HDBSCAN | 81.78 | 62.55 | -776.80 | 58.4 |
| BERTopic | 84.04 | 66.00 | -726.11 | 62.3 |
| Our Method | 44.29 | **82.69** | **-477.89** | 55.5 |
| Our Method (iter 2) | 56.83 | 77.49 | -579.48 | 55.5 |

Table 3: Evaluation results averaged for all issues. Last column shows the average of the total event count from each peak and for each issue. For LDA(Temporal), we assigned the document to its most probable topic if the probability was $\geq 0.5$.

all peaks at once for *LDA*. We include three additional baselines - *Temporal Filtering*, *HDBSCAN*, and *BERTopic* (Grootendorst, 2022). Note that *BERTopic*[3] is an off-the-shelf neural baseline for clustering documents. For methods other than ours, we do not incorporate a cluster membership module as we directly estimate the topics for all the documents in an extended temporal window of $d$ days before and after the peak ($d = 1$). Preprocessing and hyperparameter details are in Appendix C.

**Results.** Tab. 3 shows the aggregated results obtained for various methods across all the issues. For LDA (baseline), the events are estimated over a union of all the documents from every peak for an issue. We study the impact of event estimation with the temporal component by comparing LDA (baseline) and Temporal Filtering methods. We observe only a slight drop in average purity ($-3$ points) for the Temporal Filtering method. Further, Tab. 8 shows that in case of *Free Speech, Abortion, Immigration* issues, the purity scores are higher than LDA (baseline), which validates our hypothesis that adding a temporal dimension to event identification can help form coherent events.

## 4 Analysis and Discussion

### 4.1 Coverage vs Purity Trade off

We evaluate the trade-off between coverage and entity purity among the methods that take event temporality into account. We observe that LDA (Temporal) has a very high coverage with the least purity, which can be attributed to noise associated with the topic distributions. BERTopic improves over this method in both coverage, and purity measures across 11 issues. It even outperforms HDBSCAN in both the metrics. However, while BERTopic has increased coverage, it still fails to outperform our

method in terms of purity, and this can be primarily attributed to our inference mechanism that is based on generated event summaries.

To address low coverage issue from our method, we propose to run our framework for the second iteration by updating event summary embedding with the mean value of top-10 most representative document embeddings in the cluster (from the first iteration). In doing so, average coverage increased by +12.5 points across all issues, with minimal decrease of $< 5$ points in purity. Tab. 6 shows the results for each issue after the second iteration.

### 4.2 Impact of Merge/Remove Operations

We investigate the impact of removing cluster inconsistencies over the generated candidate events. For this analysis, we compare HDBSCAN with the same hyperparameters and input data as our method. We observe that average of the inter-event cosine similarity score between event-pairs, and across all issues is lesser by $0.14$ for our method. This indicates that our method achieves improved cluster separability after eliminating inconsistencies. Tab. 5 shows the report for each issue. Overall, the score is reduced, with one exception for the issue of *Corruption*. Manual inspection suggest that the increase can be due to removal of "good" clusters. An example is shown in Fig. 7.

### 4.3 KEYEVENTS ⇒ More Event Coherence

To better understand the advantages and disadvantages of our method, the authors manually annotate a small set of data samples for *Climate Change*. We test for *event coherence*, and *mapping quality* over this dataset. We define an event to be coherent if the top-K most representative documents of that event are in agreement with each other ($k = 3$). We also annotate to verify the validity of document-to-event assignments (*mapping quality*), where we check for agreement between the document and its

---
[3] https://maartengr.github.io/BERTopic

| Model | Event Coherence ↑ | Mapping Quality (Precision) ↑ |
|---|---|---|
| HDBSCAN | 84.90 | 62.27 |
| BERTopic | 85.48 | 69.87 |
| Our Method | **91.07** | **72.19** |

Table 4: Human evaluation results of our method.

respective event summary. The details about the experimental setup can be found in Appendix E.

The test is conducted across all events for our method, HDBSCAN, and BERTopic. To measure coherence, we first identify the top-K documents for an event based on their cosine similarity scores with the event centroid. In addition, we estimate *mapping quality* by judging if document pairs should be clustered together or not.

**Results.** The results of the human evaluation are shown in Tab. 4. Our method failed to generate coherent events for 5 out of the 56 cases for *Climate Change*, while BERTopic failed in 9 out of 62 cases (ignoring 3 cases where the annotator provided a label of $-1$). HDBSCAN failed in 8 out of 53 cases. Overall, the event coherence scores from BERTopic and HDBSCAN closely trail our method by a margin of approximately $-6$ points, implying that the generated events from these methods are coherent. However, considering the event purity scores, we conclude that these two methods are more noisy. In terms of mapping quality, our method outperforms HDBSCAN by a large margin. The precision score from BERTopic is better than HDBSCAN, indicating the effectiveness of BERTopic in grouping 'good' item pairs together over a small sample of randomly selected datapoints for the issue - *Climate Change*. More details in Appendix E.

### 4.4 LLM Usage and Efficiency

As temporal filtering results in an average of 55 event clusters per issue, we observe that using LLM for event summarization and cluster-merging incurs reasonable cost, as we discuss in Limitations.

## 5 Broader Impact

Our method and the resulting KEYEVENTS dataset could be useful for analyzing political discourse across different ideologies. As a simple case study, we illustrate how the portrayal of events varies for different political ideologies. We take an entity-based approach (Rashkin et al., 2016; Field and

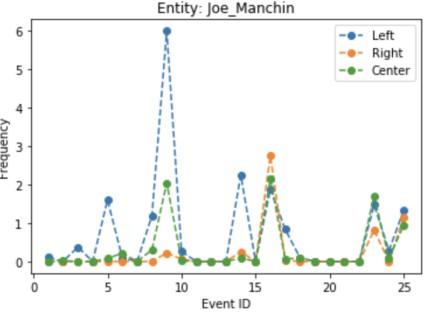

Figure 2: Frequency of the entity *Joe Manchin* (y-axis: #entity mentions per article within each event) in *Climate Change* events (x-axis: event indices across time).

Tsvetkov, 2019; Roy et al., 2021) and analyze mentions of *Joe Manchin*, a democratic senator and the chair of Senate Energy Committee, in Climate Change articles. Fig. 2 shows that left-leaning articles mention him significantly more than the other two ideologies in some of the events (e.g. the $5^{th}$, $9^{th}$, and $14^{th}$). Analyzing these events' articles show that left leaning articles criticize his ties to the coal industry and opposition to climate change legislation, while fewer (or no) mentions in articles with other ideology leanings under the same events.

Different ideologies also persist different sentiments when mentioning the same entity. In *Biden's Executive Actions on Climate Change* ($16^{th}$ event in Fig. 2), articles from different ideologies have comparable mention frequencies of *Joe Manchin*. We prompt GPT-3.5 to classify the sentiment expressed towards him (positive, neutral, negative). Interestingly, none of the articles from any ideology expresses a positive sentiment; 86% of the articles from the left endure a negative attitude towards him, whereas only 38% and 0% of the articles from the center and the right have negative sentiments. This distinction shows that even the same entities could be portrayed differently within each event to strengthen the beliefs along their political lines.

## 6 Conclusion

We present a framework for *key events* identification and showed that events generated from our approach were coherent through quantitative measures, and human evaluation. We also presented a simple qualitative study to showcase the potential of KEY EVENTS, for investigating various political perspectives under nuanced settings.

## Limitations

As the temporal filtering step of our framework relies on the publicaiton date of documents as input, we work with the assumption that the documents have a timestamp attached to them. However, the main idea of event characterization using LLM, and associating the documents to their closest event summary is applicable to other cases with no changes.

Our approach relies on GPT-3.5 for generating a multi-document event summary and cluster-merging. We choose to use GPT-3.5 instead of the open-source counterparts mostly due to computational resource constraints. Since all GPT calls are made on the cluster-level, we are able to maintain the total experimental cost of the paper under $5 with respect to the OpenAI API. To minimize the reliance and cost associated with LLM usage, we are using only pairs of documents with most similar vector representation to generate event summary. We opt for more an efficient approach here, and leave the exploration of efficiency vs. performance trade-off for future work.

## Acknowledgements

We thank the anonymous reviewers of this paper for all of their vital feedback. The project was partially funded by NSF award IIS-2135573, and in part by the Office of the Director of National Intelligence (ODNI), Intelligence Advanced Research Projects Activity (IARPA), via 2022-22072200003. The views and conclusions contained herein are those of the authors and should not be interpreted as necessarily representing the official policies, either expressed or implied, of ODNI, IARPA, or the U.S. Government. The U.S. Government is authorized to reproduce and distribute reprints for governmental purposes notwithstanding any copyright annotation therein.

## Ethics Statement

To the best of our knowledge, we did not violate any ethical code while conducting the research work described in this paper. We report the technical details needed for reproducing the results and will release the code and data collected. We make it clear that the KEY EVENTS dataset is the result of an automated algorithm not human annotation (though human evaluation was used in assessing its performance over a subset of the data).

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

## A Document Retrieval Module

This module retrieves news articles relevant to an issue of interest. User is expected to provide an issue name or a set of issue names around which the documents are to be retrieved. Using this input, we generate a set of relevant keywords associated with each issue by prompting GPT-3.5. We craft the prompt in such a way that GPT-3.5 generates a list of keywords that appear in the context of the issue specified by the user. We then use BM25 algorithm on the indexed NELA data to retrieve documents associated with each keyword for the issue. We use BM25 with the default settings for $b$, and only vary the term frequency saturation $k1 = 1.3$ as we are dealing with longer news documents.

**NELA Dataset** It is a collection of $\approx 1.8M$ news documents from 367 news outlets between January 1st 2021, and December 31st, 2021. NELA is successful in organizing the news articles based on their ideological bias. However, this structure is not well-suited to characterize the differences in discourse between the political ideologies in online news media.

In this work, we primarily focus on 207 news sources that are based out of USA. The political rating corresponding to these sources are mapped to a four-way {*left*, *right*, *center*, *conspiracy-pseudoscience*}. The ratings are decided based on MBFC [4]. Using the scores provided by MBFC, we categorize *left-center* and *right-center* political ratings to one of the {*center*, *left*, *right*} ratings.

## B Event Candidate Generation

**Temporal Filtering** We implement an outlier detection algorithm (Palshikar et al., 2009) which considers a temporal window of $2k$ points around each data point, $x$. These are $k$ points before $x$, and $k$ points after $x$. Using these $2k$ data points, we compute the mean and standard deviation. The data point is considered as a *local* peak if it is at least a standard deviation away from the mean value. Among the detected *local* peaks, we further apply a filter to retrieve *global* peaks. We do this by computing the mean and standard deviation values for the detected *local* peaks. If the value at the *local* peak is above the mean value, we mark that as a *global* peak. In the case of multiple peaks within a temporal window of $k$ days, we merge them to form a single peak. We set the value of $k = 3$ for our experiments. Figure 3 shows the result of this algorithm for the issue - *Abortion*.

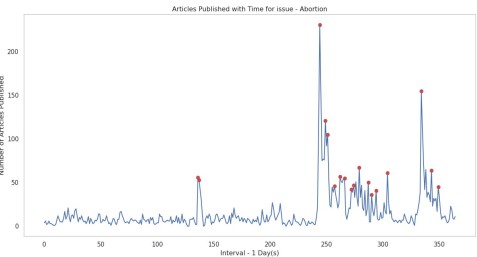

Figure 3: Dynamic Analysis of documents from Jan 1 to Dec 31, 2021, for the issue *Abortion*. $X$-axis represents time (one day interval). Red dots indicate detected peaks.

## C Models and Hyperparameters

To obtain topics from LDA with Variational Bayes sampling (under both settings), we use Gensim (Rehurek and Sojka, 2011) implementation. We follow the preprocessing steps shown in (Hoyle et al., 2021), and estimate the number of topics in a data-driven manner by maximizing . We do a grid-search over a set of $\{2, 3, 4, 5\}$ for LDA (Temporal) method. The set of topics for LDA (baseline) is $\{10, 20, \cdots, 60\}$.

---

[4]https://mediabiasfactcheck.com/

In the case of HDBSCAN, when used for our method, and as a standalone clustering model, we use a data-driven approach to estimate the best number of topics by maximizing the DBCV score (Moulavi et al., 2014). We retain the default settings for *cluster_selection_method*, and *metric* parameters, while we change the *min_cluster_size* to get more sensible topics. This number is selected based on a grid search whose values are sensitive to the number of input data points. Suppose $|X|$ denote the number of data points, then the grid parameters for HDBSCAN used in our method include $\{0.05 \times |X|, 0.06 \times |X|, \cdots 0.1 \times |X|\}$. This is updated to consider only the last three elements for HDBSCAN (standalone). If not, we see unusually high number of topics per peak. We set the *n_neighbors* parameter in UMAP embedding model to *min_cluster_size*.

For cluster incoherency check, we choose a threshold of 0.6. If the cosine similarity score between the event summary embedding and the document embedding is lower than this threshold, we discard those documents as noise.

For our method's similarity module, we choose a threshold of 0.69 based on evaluating the trade-off between purity, coherence and coverage values.

Prior to computing the TF-IDF scores to retrieve the top-K entities, we use a simple yet effective method for entity linking (Ratinov et al., 2011) that is based on Wikipedia mentions.

## D   Evaluation Metrics

In this section, we describe the evaluation metrics proposed in our work.

Several studies (Nanni et al., 2017; Spitz and Gertz, 2018; Chen et al., 2023) in the past have shown that entities and the context associated with them can potentially represent a topic or an event. With this as the premise, we have devised entity-based evaluation metrics that helps us quantify the quality of the resulting clusters. We further validate our results through a simple human evaluation process on partially annotated data for the issue - *Climate Change*.

We define **entity purity** for an event to be the proportion of the documents that are mapped to that event, where the document has at least one entity that overlaps with the top-K TF-IDF based entities for that event (k = 10). The idea is that central entities associated with a news event must be reflected in the documents clustered for that event. Note that in order to remove commonly repeated entities in news such as Biden, Trump etc., we consider top-K TF-IDF based entities for an event as central entities. A purity score of $100\%$ for an event indicates that every document in the cluster has atleast a mention of one of the top-K central entities, suggesting that each document is potentially discussing about that event.

We also define **entity coherence** metric as an additional measure to validate the cluster quality. We adapt the topic coherence metric from (Mimno et al., 2011) to define entity coherence $C$, for an event, $e_i$ as

$$C(e_i, V^{e_i}) = \sum_{m=2}^{M} \sum_{l=1}^{m-1} \log \frac{F(v_m^{e_i}, v_l^{e_i}) + \epsilon}{F(v_l^{e_i})}$$

where, $e_i$ denotes an event, $V^{e_i} = \{v_1^{e_i}, v_1^{e_i}, \cdots, v_{10}^{e_i}\}$ denotes the top-10 TF-IDF based entities for $e_i$, $F(v_m^{e_i}, v_l^{e_i})$ indicates the co-document frequency (counts the joint document frequency for entity types $v_m, v_l$), $F(v_l)$ indicates document frequency for entity type $v_l$, and $\epsilon$ is a smoothing factor. Informally, it considers co-occurrences of central entity pairs (as opposed to topic words) in the clustered documents to measure coherency for an event. Note that a higher value indicates a highly coherent news event. By virtue of using $\log$ in formula, a value closer to zero is more desirable than a largely negative value. We further observe that this measure is positively correlated with entity purity, indicating that purity can be a good measure to represent cluster coherence.

In addition to these, we have an additional metric **coverage**, which essentially counts the number of documents accounted for in the clustering process. Ideally, we want any clustering algorithm to reject noise and cluster every document in the corpus. We do not want to exclude any document. Post noise removal, a good clustering algorithm is expected to have a coverage of $100\%$ in an ideal scenario.

## E   Human Evaluation

For the **event coherence** case, the annotators are asked to verify if the top-3 documents for the event are in agreement with each other. They are asked to provide a score of $1$ if the documents are in agreement, a score $0$ if they are not, or a score of $-1$ if they are not sure about the label. We show only the *title* and *first four lines* of the news article. We did not receive any $-1$ for this case.

To evaluate the ***mapping quality*** of our model, we randomly sample a set of peaks, and within each peak, we randomly sample 50 documents to form an overall set of 430 documents mapped to various events for the issue *Climate Change*. We show the *title* and *first four lines* of news article, and the event summary to the annotators. Similar to coherence case, we ask the annotators to provide a score of 1 if the document aligns with the summary, 0 for no alignment, or −1 if they are not sure. There is no clear definition of alignment, and we let the annotators make this judgement. We received a total of 6 not sure labels. On eliminating unsure instances, our method got 352 out of 424 instances correct, which translates to a precision value of $\approx 0.83$.

However, in order to compare the performance of our method with other model, we devise a strategy to derive 'good' and 'bad' example-pairs by treating human-labeled data as the gold standard. We assume that if the two documents receive a score of 1 within the same event, then they must be 'equivalent'.

With this assumption, for a given temporal peak and within every event, we construct 'good/positive' example set by considering every possible document-pairs from valid cluster assignments. To construct 'bad/negative' example set, we consider a union of the following - (a) Document-pairs from valid cluster assignments between different events; (b) Document-pairs from an invalid, and a valid cluster assignment within each event.

The task is to evaluate how well each method performs in retaining the good example-pairs within the same cluster. We ensure to remove all the documents that are not mapped to any event by each method. Owing to the nature of data collection, we report only the precision values for all the three methods under consideration.

# F   Results

# G   Prompt Templates

This section shows the prompt templates used to generate multi-document summary (Fig. 9), and to verify if a pair of cluster characterization is equivalent (Fig. 10).

| Issue | Model | Avg. Inter-Event Cosine Similarity | # Events | # Merge Operations | # Remove Operations |
|---|---|---|---|---|---|
| Capitol Insurrection | HDBSCAN | 0.864877655 | 64 | - | - |
| | Our Method | **0.641329667** | 40 | 21 | 3 |
| Coronavirus | HDBSCAN | 0.860832152 | 122 | - | - |
| | Our Method | **0.857558543** | 112 | 10 | 2 |
| Climate Change | HDBSCAN | 0.833522985 | 74 | - | - |
| | Our Method | **0.772742185** | 56 | 11 | 7 |
| Free Speech | HDBSCAN | 0.847346069 | 72 | - | - |
| | Our Method | **0.668949583** | 56 | 7 | 13 |
| Abortion | HDBSCAN | 0.877382542 | 48 | - | - |
| | Our Method | **0.410449078** | 24 | 20 | 4 |
| Immigration | HDBSCAN | 0.852341823 | 64 | - | - |
| | Our Method | **0.75051009** | 48 | 15 | 1 |
| Gun Control | HDBSCAN | 0.829052923 | 60 | - | - |
| | Our Method | **0.663993032** | 40 | 9 | 9 |
| Criminal Injustice & Law Enforcement | HDBSCAN | 0.824876478 | 70 | - | - |
| | Our Method | **0.581169596** | 48 | 7 | 13 |
| Racial Equity | HDBSCAN | 0.839611843 | 98 | - | - |
| | Our Method | **0.730141103** | 68 | 13 | 17 |
| Defense and National Security | HDBSCAN | 0.837432569 | 106 | - | - |
| | Our Method | **0.835570683** | 89 | 11 | 6 |
| Corruption | HDBSCAN | 0.818098607 | 46 | - | - |
| | Our Method | 0.821913246 | 30 | 5 | 31 |

Table 5: Shows the impact of cluster merge and remove operations for each issue. Note that input data and hyper-parameters used by HDBSCAN in this setting are the same as our method. Lower the similarity score the better.

| Issue | Coverage | Avg. Entity Purity | Avg. Entity Coherence |
|---|---|---|---|
| Capitol Insurrection | 65.164 | 72.253 | -619.226 |
| Coronavirus | 54.562 | 56.159 | -762.147 |
| Climate Change | 56.263 | 84.509 | -519.816 |
| Free Speech | 47.378 | 78.124 | -589.486 |
| Abortion | 87.946 | 66.658 | -739.842 |
| Immigration | 67.398 | 76.275 | -618.273 |
| Gun Control | 46.797 | 88.781 | -427.161 |
| Criminal Injustice & Law Enforcement | 38.701 | 87.209 | -561.958 |
| Racial Equity | 41.966 | 82.548 | -549.100 |
| Defense and National Security | 55.264 | 84.907 | -451.647 |
| Corruption | 63.702 | 79.943 | -535.650 |
| **Average Stats** | **56.831** | **77.942** | **-579.482** |

Table 6: Statistics for our proposed method after increased coverage with acceptable reduction in entity purity ($\approx -5$ points on an average for all issues).

| Summary | Document |
|---|---|
| | *Vice President Pence supports Congress members who will object to Biden's designation Jan. 6.* |
| *News Event Title: Election Fraud Claims in the US. News Event Description: This is about the claims of election fraud in the US and the upcoming congressional meeting to certify the Electoral College votes.* | *Vice President Mike Pence welcomed the decision by a group of senators, led by Sen. Ted Cruz ( R-Texas ), to challenge the scheduled nomination of Democratic presidential candidate Joe Biden as the winner of the election held on Nov. 3. The vice president welcomes the efforts of members of the House and Senate to use the authority they have under the law to raise objections and bring forward evidence before the Congress and the American people, Pence's chief of staff Marc Short said, according to Axios on Jan. 3.@ @ @ @ @ @ of millions of Americans about voter fraud and irregularities.* |

Table 7: Illustrates an example where the cluster was removed due to the document being present in the top-5 list for this event. We see that the document is talking about the same issue from a different frame and merely, using a similarity module to identify cluster incoherency is not sufficient in this case.

| Issue | Model | Coverage | Avg. Entity Purity | Avg. Entity Coherence | Agg. Event Count |
|---|---|---|---|---|---|
| **Capitol Insurrection** | LDA (baseline) | 99.781 | 36.058 | -1027.214 | 60 |
| | Temporal Filtering | - | 27.867 | -1092.882 | 17 |
| | LDA (Temporal) | 85.491 | 37.129 | -1025.687 | 64 |
| | HDBSCAN (standalone) | 77.964 | 54.155 | -888.38 | 50 |
| | BERTopic | 83.351 | 64.819 | -791.722 | 54 |
| | Our Method | 47.349 | **76.821** | **-547.06** | 40 |
| **Coronavirus** | LDA (baseline) | 99.774 | 17.885 | -1003.54 | 60 |
| | Temporal Filtering | - | 8.79 | -1184.476 | 21 |
| | LDA (Temporal) | 62.784 | 14.487 | -1110.409 | 83 |
| | HDBSCAN (standalone) | 65.586 | 34.458 | -1004.468 | 64 |
| | BERTopic | 61.731 | 35.915 | -941.667 | 54 |
| | Our Method | 41.965 | **56.299** | **-749.045** | 112 |
| **Climate Change** | LDA (baseline) | 99.767 | 42.439 | -883.566 | 60 |
| | Temporal Filtering | - | 28.02 | -1040.555 | 18 |
| | LDA (Temporal) | 90.89 | 39.806 | -957.687 | 64 |
| | HDBSCAN (standalone) | 84.011 | 64.148 | -763.608 | 53 |
| | BERTopic | 83.595 | 67.635 | -689.429 | 65 |
| | Our Method | 45.015 | **81.528** | **-453.923** | 56 |
| **Free Speech** | LDA (baseline) | 99.684 | 21.785 | -1090.102 | 60 |
| | Temporal Filtering | - | 30.039 | -1105.5 | 20 |
| | LDA (Temporal) | 93.135 | 41.441 | -1032.338 | 68 |
| | HDBSCAN (standalone) | 83.175 | 65.337 | -772.847 | 72 |
| | BERTopic | 83.649 | 70.303 | -704.514 | 75 |
| | Our Method | 35.46 | **87.964** | **-439.135** | 56 |
| **Abortion** | LDA (baseline) | 99.078 | 33.739 | -917.643 | 60 |
| | Temporal Filtering | - | 36.691 | -1045.857 | 14 |
| | LDA (Temporal) | 93.436 | 48.161 | -914.619 | 48 |
| | HDBSCAN (standalone) | 79.04 | 70.162 | -732.593 | 37 |
| | BERTopic | 85.655 | **71.765** | -733.281 | 42 |
| | Our Method | 77.198 | 70.332 | **-594.95** | 24 |
| **Immigration** | LDA (baseline) | 99.746 | 24.253 | -1033.2 | 60 |
| | Temporal Filtering | - | 24.781 | -1060.21 | 19 |
| | LDA (Temporal) | 87.848 | 34.72 | -993.803 | 66 |
| | HDBSCAN (standalone) | 79.944 | 61.818 | -776.407 | 54 |
| | BERTopic | 86.339 | 67.634 | -713.125 | 56 |
| | Our Method | 53.964 | **80.107** | **-535.755** | 48 |
| **Gun Control** | LDA (baseline) | 99.606 | 26.002 | -1049.5 | 60 |
| | Temporal Filtering | - | 35.109 | -903.333 | 18 |
| | LDA (Temporal) | 90.146 | 42.534 | -955.083 | 61 |
| | HDBSCAN (standalone) | 91.494 | 67.047 | -649.708 | 48 |
| | BERTopic | 94.906 | 66.774 | -675.880 | 50 |
| | Our Method | 36.306 | **95.124** | **-323** | 40 |
| **Criminal Injustice & Law Enforcement** | LDA (baseline) | 99.85 | 40.432 | -996.468 | 60 |
| | Temporal Filtering | - | 31.152 | -1075.809 | 20 |
| | LDA (Temporal) | 96.648 | 45.199 | -1027.712 | 66 |
| | HDBSCAN (standalone) | 87.968 | 67.118 | -796.317 | 68 |
| | BERTopic | 88.725 | 67.105 | -756.769 | 78 |
| | Our Method | 31.368 | **94.194** | **-463.652** | 48 |
| **Racial Equity** | LDA (baseline) | 99.79 | 31.377 | -1073 | 60 |
| | Temporal Filtering | - | 30.931 | -1109.25 | 24 |
| | LDA (Temporal) | 93.893 | 40.448 | -1040.695 | 82 |
| | HDBSCAN (standalone) | 80.344 | 63.346 | -811.065 | 76 |
| | BERTopic | 85.374 | 66.614 | -747.699 | 75 |
| | Our Method | 33.206 | **89.082** | **-369.184** | 68 |
| **Defense & National Security** | LDA (baseline) | 99.951 | 38.158 | -940.564 | 60 |
| | Temporal Filtering | - | 25.312 | -1098.041 | 24 |
| | LDA (Temporal) | 91.609 | 40.008 | -1008.138 | 87 |
| | HDBSCAN (standalone) | 84.319 | 71.648 | -686.023 | 84 |
| | BERTopic | 89.004 | 74.519 | -617.425 | 87 |
| | Our Method | 40.083 | **90.61** | **-353.291** | 89 |
| **Corruption** | LDA (baseline) | 99.572 | 34.557 | -1023.875 | 60 |
| | Temporal Filtering | - | 30.965 | -961.727 | 11 |
| | LDA (Temporal) | 93.33 | 40.925 | -992.941 | 34 |
| | HDBSCAN (standalone) | 85.763 | 68.762 | -663.4 | 36 |
| | BERTopic | 82.115 | 73.368 | -615.773 | 50 |
| | Our Method | 45.233 | **87.577** | **-427.75** | 30 |

Table 8: Compares the results obtained for each method and issue. Last column shows summation of all event counts (from each detected temporal peak). For LDA(Temporal), we assigned the document to its most probable topic if the probability was $\geq 0.5$.

You need to provide a title and a sentence long description for the news event based on news article snippets shown below. The title and description should not be too specific to the articles shown below but rather, they need to focus on the main event.

News Article1: **Title**
*Description**

News Article2: **Title**
*Description**

News Event Title: **Response**
News Event Description: **Response**

News Article1: **Title**
*Description**

News Article2: **Title**
*Description**

Table 9: Prompt template for multi-document event summary generation (shown as one-shot).

You need to tell if the following two news event descriptions belong to the same news event. You need to say yes or no and nothing more.

News Event Title1: **Title**
*Description**

News Event Title2: **Title**
*Description**

Answer:

Table 10: Prompt template to check for entailment (shown as zero-shot).