# OpenReview forum: "Using LLM for Improving Key Event Discovery: Temporal-Guided News Stream Clustering with Event Summaries"
_EMNLP/2023/Conference — EMNLP 2023 Findings_

### Official Review · Reviewer_9es6 · 2023-08-04

**Soundness:** 4

**Excitement:**

3: Ambivalent: It has merits (e.g., it reports state-of-the-art results, the idea is nice), but there are key weaknesses (e.g., it describes incremental work), and it can significantly benefit from another round of revision. However, I won't object to accepting it if my co-reviewers champion it.

**Missing References:**

-

**Paper Topic And Main Contributions:**

The paper considers the task of topical event extraction from news, for which the authors propose a a neural pipeline that combines large language models with outlier detection. The proposed approach is evaluated against topic models as a baseline.

**Questions For The Authors:**

Less of a question and more of a comment:
In the discussion of limitations, you argue for using GPT 3.5 and against open models, which rather feels like a statement made out of convenience. Yes, using a nicely curated OpenAI API is easy, but it has inherent limitations in applicability, suffers from privacy cocerns, and offers no control over the model. Adding support for an open model would not only allow further and more in-depth experiments, but also make the final produc more accessible.

**Reasons To Accept:**

* Presentation
The paper is well written and the proposed approach is presented clearly, with extensive information regarding the implementation being provided in the appendix.

* Usefulness
While the proposed pipeline is of limited novelty and based on the combination of existing solutions and concepts, the final product is of good quality and could be useful to the community. Providing a useable code repository / package would be even more beneficial.

* Data
The authors provide a data set alongside their experiments.

**Reasons To Reject:**

* Limited novelty
The proposed approach is a effectively a combination of existing tools and well-established ideas. Novelty is only in the engineering / data, not in the methods.

* Limited evaluation
The used baselines are quite outdated, to the point where the comparison is unfair (LDA is 20 years old at this point). It would be good to see the relative performance of more recent neural topic models or best-effort adaptations of LLMs to the task.

**Reproducibility:**

4: Could mostly reproduce the results, but there may be some variation because of sample variance or minor variations in their interpretation of the protocol or method.

**Reviewer Confidence:**

4: Quite sure. I tried to check the important points carefully. It's unlikely, though conceivable, that I missed something that should affect my ratings.

**Typos Grammar Style And Presentation Improvements:**

The paper is very well written and enjoyable to read with a nice overall presentation. Great job!

Typos:
* Line 244: "We observe that average"
* Line 255: "results in on average"

---

> ### Author Rebuttal · Authors · 2023-08-29
>
> Thank you for your comments!
>
> 1. Limited novelty The proposed approach is a effectively a combination of existing tools and well-established ideas. Novelty is only in the engineering / data, not in the methods.
>
> While clustering approaches - probabilistic or neural methods is an established idea, the resulting clusters are still far from perfect. Our main goal and novelty in this paper is to come up with a principled way of using LLM’s capability as an inference engine to help improve the overall cluster quality.
>
> Precisely, post temporal filtering, we identify the event candidates by clustering the documents using HDBSCAN algorithm. Each event candidate is characterized with a short explanation using LLM. This characterization is utilized to remove incoherent events, and also to merge similar events descriptions together. In the inference stage, we use the LLM-generated event characterization to map the documents to the event cluster.
>
> As far as our knowledge is concerned, we do not know of anybody who has used LLM’s inference capability in this manner to generate coherent news event clusters. Our empirical results show the potential in our approach.
>
>
> 2. Limited evaluation The used baselines are quite outdated, to the point where the comparison is unfair (LDA is 20 years old at this point). It would be good to see the relative performance of more recent neural topic models or best-effort adaptations of LLMs to the task.
>
> Thanks for suggesting that we compare our proposed approach against recent neural topic models. We have compared our method against BERTopic, and will be reporting full results in the final version of the paper. In fact, keeping the temporal clustering aside, our current baseline - HDBSCAN (standalone), can be considered as a reimplementation of BERTopic. This is because we first obtain a distributed representation of documents, then reduce the dimensionality using UMAP, and finally, apply HDBSCAN algorithm to retrieve news events.
>
> On using the ‘off-the-shelf’ model - BERTopic for key event identification, we found that it outperformed HDBSCAN (standalone) baseline in terms of coverage, and entity purity metrics. Coverage increased by +3 points, and entity purity increased by +3.5 points. The reason for this variation can be primarily attributed to the change in hyperparameters (in UMAP and HDBSCAN algorithm) underlying the readily available implementation of BERTopic. This change in parameters led the stochastic clustering process to create more events (+4 events on average across all issues), and help reduce the noise in event clusters.
>
> However, our method produced a lesser number of events overall due to the merge/remove operations on event candidates, and also, entity purity scores are atleast +11 points higher than BERTopic. The increase in the purity metric can be directly attributed to our additional inference mechanism that is solely based on generated event summaries. This inference mechanism is absent in clustering approaches used in baselines.
>
> We have also conducted human evaluation on results from BERTopic baseline, and will be added as part of Table 4 in the paper. Event Coherence and Mapping Quality scores from BERTopic increased by +0.5 points and +7.5 points respectively, as compared to baseline - HDBSCAN (standalone). The scores from BERTopic are still lesser than our method by -5.5 points and -2.3 points for event coherence and mapping quality, respectively. After combining these observations along with the reduced overall purity scores from BERTopic, we can conclude that events produced by BERTopic are still prone to noise compared to our method.
>
>
> 3. Less of a question and more of a comment: In the discussion of limitations, you argue for using GPT 3.5 and against open models, which rather feels like a statement made out of convenience. Yes, using a nicely curated OpenAI API is easy, but it has inherent limitations in applicability, suffers from privacy cocerns, and offers no control over the model. Adding support for an open model would not only allow further and more in-depth experiments, but also make the final produc more accessible.
>
> We agree with your point. It is indeed a statement made out of convenience – With the limited computational resource we had for this project, it was more difficult to setup an open-source model (e.g. LLaMA-7B) compared to using GPT3.5 (which didn’t cost much in our experiments) for inference. In the final draft, we will try to add an ablation study for using open-source LLMs, just so we understand how much performance we will lose/gain when switching to open-source counterparts.
>
>
> 4. Typos
>
> Thanks for pointing this out. We will fix them in the final draft.

---

### Official Review · Reviewer_DScZ · 2023-08-10

**Soundness:** 3

**Excitement:**

3: Ambivalent: It has merits (e.g., it reports state-of-the-art results, the idea is nice), but there are key weaknesses (e.g., it describes incremental work), and it can significantly benefit from another round of revision. However, I won't object to accepting it if my co-reviewers champion it.

**Paper Topic And Main Contributions:**

this paper studies how to identify key events and news articles related to these events from the news flow. In this paper, a general framework of news stream clustering is proposed, which makes use of the time trend of news. LLM generates event summaries to describe such key events, and on this basis, document clusters are formed in an unsupervised way. The framework is effective after a series of evaluations and shows that it produces a more coherent event cluster.

**Reasons To Accept:**

An efficient method is proposed for a novel task, and experimental results show that it is computationally inexpensive and produces more interpretable coherent clusters of key events.

**Reasons To Reject:**

The motivation in this article is not elaborated, the most interesting is the peak, but with a relevant simple operation.

It is not clear how event clustering is implemented. What is the form of events contained in a peak, what is the connection with the passage, and how to obtain the summary, it is recommended to reorganize.

The evaluation indicators can be explained in more detail. This article uses multiple evaluation indicators, how they are defined, and what they represent. I hope they can be supplemented.

The image layout in the appendix is confusing.

**Reproducibility:**

3: Could reproduce the results with some difficulty. The settings of parameters are underspecified or subjectively determined; the training/evaluation data are not widely available.

**Reviewer Confidence:**

4: Quite sure. I tried to check the important points carefully. It's unlikely, though conceivable, that I missed something that should affect my ratings.

---

> ### Author Rebuttal · Authors · 2023-08-29
>
> Thank you for your comments!
>
> 1. The motivation in this article is not elaborated, the most interesting is the peak, but with a relevant simple operation.
>
> Our goal is to retrieve major news events from a large corpus of news articles with minimum human intervention. The event identification information derived from temporal clustering alone is very ambiguous. This is because of the presence of multiple news events that are often conflated around every temporal peak.
>
> Our main challenge evolves to come up with a principled approach to identify news events at every peak. To tackle this problem, we propose a framework to effectively cluster the news articles into various events at each peak by using LLM for inference purposes. Specifically, we propose to use LLM to characterize and explain the identified event candidates, and then clean up the candidate events through merge/remove operations. The generated event explanations of the final set of clusters are utilized to decide what documents are mapped to each event cluster.
>
>
> 2. It is not clear how event clustering is implemented. What is the form of events contained in a peak, what is the connection with the passage, and how to obtain the summary, it is recommended to reorganize.
>
> - Event Clustering (Main idea): Post application of temporal clustering, we use HDBSCAN algorithm over a distributed representation of documents to identify event candidates. We prompt LLM to generate an explanation for each event candidate. This explanation is utilized to purify candidate events through remove/merge operations. The resulting output after purification of event candidates denotes the final set of event clusters. LLM-generated explanations associated with the final set of event clusters are used to retrieve news articles related to each news event.
>
> - Form of events: Intuitively, this can be thought of as a theme extraction process. We have a corpus of news articles at different points in time (peak). At each peak, we are to identify various points of discussion that constitute a major chunk of news articles, which we coin as “news events”. Each news event has a title and a brief description that tells what the event is about, and each event is associated with a collection of news articles.
>
> - Connection with passage and Obtaining Summary: To obtain an event summary, the identified event candidates from HDBSCAN algorithm are characterized using LLM. Specifically, we retrieve the top-3 representative news articles from each event candidate based on cosine similarity score between event centroid and document embeddings for that event. To obtain an event summary, we prompt the LLM with the news article title and first four lines in a few-shot manner to generate a short explanation about that news event.
>
>
> 3. The evaluation indicators can be explained in more detail. This article uses multiple evaluation indicators, how they are defined, and what they represent. I hope they can be supplemented.
>
> We can certainly add more details about the evaluation metrics in the appendix section of the final draft. This can perhaps help in explaining them better. We provide a brief overview here.
>
> - Broadly, we have two kinds of evaluation - human evaluation, and other measures that are based on traditional clustering evaluation metrics.
>
> - Human Evaluation: We have evaluation indicators - Event Coherence and Mapping Quality that are based on human judgements, and can be found in Table 4 in the paper. Event Coherence for an event is measured by checking if the top-3 documents from each event cluster are in agreement with each other, whereas Mapping Quality models if the contents of the news article align with the event summary. Note that there is no clear definition of alignment and we let the annotators make this judgement.
>
> - Other measures: The other indicators are based on traditional clustering-related evaluation metrics that considers coherence for an event (Table 3) and separability across different clusters (Table 6). For instance, to quantify the coherence of the identified events, we propose entity purity. The motivation underlying this metric is that central entities associated with a news event must be repeated in the documents associated with that event. A purity score of 100% for an event indicates that every document in the cluster has atleast a mention of one of the top-K central entities, suggesting that each document is potentially discussing that event.
>
>
> 4. The image layout in the appendix is confusing.
>
> Thank you. We will fix this issue in the final draft of our paper.

---

### Official Review · Reviewer_RPBd · 2023-08-12

**Soundness:** 4

**Excitement:**

3: Ambivalent: It has merits (e.g., it reports state-of-the-art results, the idea is nice), but there are key weaknesses (e.g., it describes incremental work), and it can significantly benefit from another round of revision. However, I won't object to accepting it if my co-reviewers champion it.

**Paper Topic And Main Contributions:**

This paper is about generating key event aggregation from news articles, specifically NELA. The authors try to cluster different news articles based on specific topics for better analysis of social media news understanding and propagation. The authors have used GPT 3.5 with prompt assistance to further overcome the challenge of under-specificity of social media news articles. They use different merge and remove operations to generate consistent clusters, ultimately mapping articles to events.

**Questions For The Authors:**

(a)  Why only HDBSCAN is used for clustering?
(b)  Why other methods of dimensionality reduction except UMAP not used?
(c)   Why only NELA dataset is used?

**Reasons To Accept:**

This work seems to be novel and the first of its kind that generates fine grained and consistent clusters of news articles. The objective is well motivated and experiments seem to justify the author's claims. The authors have used different metrics like purity, coverage and also human evaluation to benchmark their experiments. Different baselines have been compared with and the proposed method seems to outperform all baselines. The dataset 'KEYEVENTS' proposed by the authors would help the research community to steer research in social media news analysis.

**Reasons To Reject:**

This paper could be improved by exploring recent state of the art topic modelling models such as TopicBert can be explored.

**Reproducibility:**

3: Could reproduce the results with some difficulty. The settings of parameters are underspecified or subjectively determined; the training/evaluation data are not widely available.

**Reviewer Confidence:**

3: Pretty sure, but there's a chance I missed something. Although I have a good feel for this area in general, I did not carefully check the paper's details, e.g., the math, experimental design, or novelty.

---

> ### Author Rebuttal · Authors · 2023-08-29
>
> Thank you for your comments!
>
> 1. This paper could be improved by exploring recent state of the art topic modelling models such as TopicBert can be explored.
>
> Thanks for suggesting that we compare BERTopic (SBERT + UMAP + HDBSCAN) as an added baseline. We have compared our method against it, and will be reporting full results in the final version of the paper. In fact, keeping the temporal clustering aside, our current baseline - HDBSCAN (standalone), can be considered as a reimplementation of BERTopic. This is because we first obtain a distributed representation of documents, then reduce the dimensionality using UMAP, and finally, apply HDBSCAN algorithm to retrieve news events.
>
> We found that off-the-shelf BERTopic outperformed HDBSCAN (standalone) baseline in terms of coverage (+3 points), and entity purity metrics (+3.5 points). The reason for this variation can be primarily attributed to the change in hyperparameters (in UMAP and HDBSCAN algorithm) underlying the readily available implementation of BERTopic. This change in parameters led the stochastic clustering process to create more events (+4 events on average across all issues), and help reduce the noise in event clusters.
>
> Our proposed method outperforms both BERTopic and HDBSCAN baselines by a sizable margin. The most notable improvements from our methods are – (1) our method produces a lesser number of events overall due to the merge/remove operations on event candidates. The increase in the purity metric (+11 points vs. BERTopic) can be directly attributed to our additional inference mechanism that is solely based on generated event summaries. This inference mechanism is absent in clustering approaches used in baselines.
>
> We have also conducted human evaluation on results from BERTopic baseline, and will be added as part of Table 4 in the paper. Event Coherence and Mapping Quality scores from BERTopic increased by +0.5 points and +7.5 points respectively, as compared to baseline - HDBSCAN (standalone). The scores from BERTopic are still lesser than our method by -5.5 points and -2.3 points for event coherence and mapping quality, respectively. After combining these observations along with the reduced overall purity scores from BERTopic, we can conclude that events produced by BERTopic are still prone to noise compared to our method.
>
>
>
> 2. Why only HDBSCAN is used for clustering?
>
> The idea is to use a density-based estimate such as HDBSCAN to retrieve an event representation by estimating areas of highly similar documents in the semantic space. Further, given that this is a short paper, it was out-of-scope for us to compare various clustering approaches.
>
>
>
> 3. Why other methods of dimensionality reduction except UMAP not used?
>
> We have tried using other popular dimensionality reduction techniques such as t-SNE, and PCA. However, we chose to go with UMAP as it preserves both local and global structure in the data, which makes it very useful for us to compare inter-cluster distances. Furthermore, it is known that UMAP scales well for large datasets in comparison to t-SNE, which incentivized us to use UMAP for our use case.
>
>
>
> 4. Why only NELA dataset is used?
>
> We have used NELA dataset-2021 as it is an accumulation of a large collection of news articles (~1.8 Million) from 367 different news sources for the year 2021.  We have leveraged it to illustrate our conceptual idea of using LLMs as an inference engine to retrieve better clustering results. However, in practice we can leverage this approach to identify themes in any large corpus of news articles.

---

### Meta-Review · Area_Chair_6PdX · 2023-09-23

**Recommendation:** 4

**Metareview:**

This paper proposes a method to cluster news articles into key events using a large language model (LLM) and outlier detection. The paper uses the NELA dataset of news articles from various sources and domains. The paper first uses the LLM to generate event summaries from the articles and then remove irrelevant or noisy articles from the clusters. The authors evaluate the proposed method against topic models as baselines and shows that it produces more coherent and informative event clusters.

I thank the reviewers for providing very constructive feedback and the authors for addressing concerns by executing additional experiments. Especially, the inclusion of BERTopic as an additional baseline is commendable. I would encourage the authors to consider other suggestions such as using another LLM such as Llama2 in addition to GPT3.5 model currently used.

---

### Decision · Program_Chairs · 2023-10-07

**Decision:**

Accept-Findings

**Comment:**

This paper proposes a method to cluster news articles into key events using a large language model (LLM) and outlier detection. The paper uses the NELA dataset of news articles from various sources and domains. The paper first uses the LLM to generate event summaries from the articles and then remove irrelevant or noisy articles from the clusters. The authors evaluate the proposed method against topic models as baselines and shows that it produces more coherent and informative event clusters.

I thank the reviewers for providing very constructive feedback and the authors for addressing concerns by executing additional experiments. Especially, the inclusion of BERTopic as an additional baseline is commendable. I would encourage the authors to consider other suggestions such as using another LLM such as Llama2 in addition to GPT3.5 model currently used.